# Did Harvey Learn from Katrina? Initial Observations of the Response to Companion Animals during Hurricane Harvey

**DOI:** 10.3390/ani8040047

**Published:** 2018-03-30

**Authors:** Steve Glassey

**Affiliations:** Public Safety Institute of New Zealand, P.O. Box 216, Wellington 6140, New Zealand; steve@publicsafety.institute; Tel.: +64-210-278-8930

**Keywords:** animals, disasters, hoarding, Hurricane Harvey

## Abstract

**Simple Summary:**

When Hurricane Harvey struck the Gulf states in 2017, a large-scale rescue effort was launched by officials and citizens to rescue both people and animals. Over a decade since Hurricane Katrina (2005), this study explores whether the reforms to afford better protection to companion animals such as the Pet Emergency and Transportation Standards Act 2006 have made a difference. Key officials from various organizations within the state of Texas were interviewed and it was found that though there has been a cultural shift to better protect animals in a disaster, formal coordination and planning mechanisms need further attention. This study also uncovered the first empirical observation of disaster hoarding where such persons used the disaster to replenish their animal stocks. This study will be of interest to those involved in emergency management and animal welfare.

**Abstract:**

The aftermath of Hurricane Katrina in 2005 became the genesis of animal emergency management and created significant reforms in the US particularly the passage of the Pets Emergency and Transportation Standards Act in 2006 that required state and local emergency management arrangements to be pet- and service animal-inclusive. More than a decade later Hurricane Harvey struck the Gulf states with all 68 directly related deaths occurring in the state of Texas. In this study, six key officials involved in the response underwent a semi-structured interview to investigate the impact of the PETS Act on preparedness and response. Though the results have limitations due to the low sample size, it was found that the PETS Act and the lessons of Hurricane Katrina had contributed to a positive cultural shift to including pets (companion animals) in emergency response. However, there was a general theme that plans required under the PETS Act were under-developed and many of the animal response lessons from previous emergencies remain unresolved. The study also observed the first empirical case of disaster hoarding which highlights the need for animal law enforcement agencies to be active in emergency response.

## 1. Introduction

In 2005, Hurricane Katrina struck the United States Gulf Coast causing more than 1245 human deaths and, at the time, was the costliest disaster in US history (2017: USD$161.3 billion). This historic event epitomised the plight of animals being vulnerable to disaster and the strong bond many animal owners had with their pets. Forty-four percent of those choosing not to evacuate doing so, in part, because they were unable to take their pets with them [1]. This experience led to major reforms including the introduction of the Pets Emergency and Transportation Standards (PETS) Act of 2006 that required state and local plans to ensure the needs of companion and service animals were met in future planning and operations. Twelve years later, Hurricane Harvey struck the Gulf States affecting over 330,000 structures, flooding over 500,000 vehicles and forcing over 40,000 people to be housed in emergency accommodation [2]. There were particularly disastrous consequences for the city and surrounds of Houston with over half the human deaths recorded in Harris County and City of Houston [2]. Hurricane Harvey will become the second-costliest disaster in US history (once adjusted for inflation) with an estimated bill over of USD$125 billion in damages [2]. The images of Hurricane Harvey across global media documented one of community-centric response with officials calling for anyone with a boat to help assist with flood-related rescues, the Cajun Navy responding to such calls, and a significant effort to ensure the errors of Hurricane Katrina, including leaving pets behind would not happen again. 

The Federal Emergency Management Agency (FEMA) estimated over 30,000 rescues were carried out during Hurricane Harvey. Despite the damage, Hurricane Harvey had a considerably lower death toll of at least 68 human lives lost directly as a result in Texas, the largest number of direct deaths from a tropical cyclone in that state since 1919 [2]. The State of Texas is the second largest State in the US with an estimated 27,862,596 residents across its 254 counties; including 4,589,928 in Harris County, which is home to the fourth largest city in the US, Houston [3].

There are many differences between the two Hurricanes and why their damages and fatalities may be so contrasting; however, the aim of this article is to explore Hurricane Harvey as a critical case study to qualitatively evaluate the effectiveness of response in a post-PETS Act implementation era. The significance of this study will help inform improvements to animal emergency management practices which ultimately influence not only protection of animals, but humans as well who are likely to exhibit protective behaviours as a result of the human-animal bond [4].

Hurricane Katrina generated a significant amount of empirical disaster related research across a wide range of scholarly disciplines. However, due to significant changes following this event such as legal reforms, this research and its recommendations may no longer provide current advice.

## 2. Method

A field visit was undertaken between 19 and 22 December 2017 to conduct pre-arranged meetings with those who were significantly involved in leading animal-inclusive emergency responses to Hurricane Harvey in the State of Texas. A semi-structured interview was undertaken which took between one hour and three and a half hours depending on the time available subjects had. Subjects were chosen for their organisation’s high public profile through social media activity during the response to Hurricane Harvey and to ensure a cross-section of not-for-profit and local government officials who held a leadership role were interviewed. Six subjects were interviewed, and all received follow-up emails to clarify notes taken and request feedback on the final manuscript prior to peer review. To supplement the interviews, a cursory analysis of online traditional media articles was also undertaken. The interview focused on four key research questions: (1) Had the PETS Act 2006 influenced animal emergency management practices? (2) What preparatory activities had been undertaken to protect animals prior to Hurricane Harvey? (3) What were the challenges and novel complications observed by those leading the animal emergency response to Hurricane Harvey? (4) What were the key lessons from Hurricane Harvey, from an animal emergency management perspective? The semi-structured nature of the interview allowed other areas to be discussed and documented in the interview notes to provide clarity over issues raised by the respondent. Contact was made with various government organizations, such as the Texas Animal Health Commission and Harris County Public Health to validate claims made mainly regarding lack of planning and coordination. The method employed, however, does have its limitations given the small sample size and that the interview subjects are likely to exhibit a positive bias toward animal welfare given their organizations purpose and/or pro-animal welfare individual comments made in the media. The sample group also, in most cases, were active in animal welfare, and a study by Taylor et al. [5] found such groups were more likely to report issues with animal emergency response than mainstream emergency service organizations. A positive bias toward animal welfare may also be typical of modern society, where many have companion animals and see these as members of their family [6,7].

## 3. Preliminary Results and Discussion

### 3.1. Impact of the PETS Act 2006

In discussion with the interviewed respondents, it was clear that only a minority had specific knowledge of the PETS Act. Those that did have knowledge of it displayed disappointment that it was more tokenism, lacked implementation at a practical level, and was characterised as “no carrot and no stick”. However, it was unanimous across the respondents that the memories of Hurricane Katrina had shaped a culture where the evacuation of companion animals alongside their human counterparts was now a cultural norm. This was consistent with the findings by Hunt, Bogue, and Rohrbaugh [8]. At the state level, there was no animal emergency plan in effect according to most respondents, though a draft was under development. At county and city levels, there were also no animal emergency plans in effect known to the respondents, however, a draft coordination document for Harris County was later supplied. Neither state nor local animal emergency plans were available publicly online. These observations contrast with the requirements under the PETS Act for state and local plans to include companion and service animal provisions. This potentially exposes a major shortfall in the US animal emergency management environment. Though planning expectations had not been met, some areas had activity facilitating animal emergency management meetings as part of the preparedness phase such as the Harris County Disaster Animal Management Task Force which was not known to any of the respondents. Texas is unique in the sense that it has 254 counties (the largest number of counties in a US state and the next layer of administration is at the state level, leaving the state with an unreasonable span of coordination. It was clear in discussions that planning and pre-event coordination efforts were sub-optimal in Texas, but this non-compliance is not unique to Texas, with studies in other states observing similar deficiencies in planning in the past [9]. Despite the PETS Act being in place for over a decade, it appears there has been no critical evaluation of its effectiveness by the government.

### 3.2. Preparedness

In addition to the lack of state and local animal emergency plans being in effect, none of the organizations interviewed had carried out any animal emergency management training or exercises. Some humane investigators, however, had completed professional development courses in swiftwater rescue as provided by Code 3. However, it was clear that operational responders across the animal welfare groups would fall well below expectations set under the National Incident Management System’s credentialing system for animal rescue related roles. None of the animal welfare groups had a dedicated emergency manager which is typical, given the constraints that these charities operate within. However, the National Alliance of State Animal and Agricultural Emergency Programs (NASAAEP) and National Animal Rescue and Sheltering Coalition (NARSC) have developed substantial resources to assist in animal disaster preparedness.

### 3.3. Response

Following the hurricane warning, the two major animal welfare charities emptied their shelters of animals and relocated these animals to safer locations (except for animals in protective custody due to legal reasons). This allowed the shelter to be at minimal risk from the Hurricane, but also provided more capacity to shelter displaced animals after the hurricane made an impact later. The emptying of animal shelters pre- and post-impact has become a common practice in disasters, which is encouraging. 

At a local level, it appeared that animal welfare charities and emergency management operated without any significant state or county coordination or leadership. In Houston, the two major animal welfare charities appeared to already have their own operational areas with spontaneous and other groups coming from outside the area to fill coverage voids. In some disaster-affected areas in Texas, where there was no local Society for the Prevention of Cruelty to Animals (SPCA) or Humane Society, the local animal control provided a default animal emergency response service.

It was clear that in Houston there were tensions with external organizations from outside the area, let it be within the state or outside. It was alleged that some of these organizations were unfamiliar with the local veterinary challenges, such as heartworm, distemper, and parvovirus, with such organizations having outbreaks of these diseases in their temporary animal shelters. Due to the climate and other conditions, heartworm is common in Gulf States, including Texas [10], and the local animal welfare charity shelters run regular heartworm testing and dosing clinics. Some volunteers came from areas outside of Texas where heartworm was rare. These volunteers assumed that animals presenting with heartworm was a sign of neglect and their attitudes toward owners created challenges. Self-appointed journalists also reported accusations that one of the major animal groups was euthanizing flood-displaced animals [11], however, this was strongly denied, and no evidence was found to substantiate such a claim.

The US military were highly praised by all the subjects for their response efforts, including the rescuing of animals. The military responders were well-equipped with water safety equipment (Figure 1) and placed importance on the need to evacuate animals alongside their human guardians. It was also common for each military high clearance vehicle to be assigned a rescue swimmer who had specialist flood rescue equipment and training. This contrasts with the experiences in New Zealand, such as the Edgecumbe Flooding, where the New Zealand Army responded to a flooding event without any basic protective equipment, such as helmets, gloves, or personal floatation devices, and hindered aspects of the animal rescue operation [12].

In one small city, the Department of Homeland Security (DHS) deployed an Urban Search and Rescue (USAR) team to clear the flood-affected residential area. The city’s emergency manager was not consulted over their self-deployment and the team was escorted around by DHS officials overtly armed with automatic rifles. This was not well received by the local community. The team applied the FEMA search marking system, however in direct observation as provided through a tour of the flood-affected area, those markings appeared to be incomplete. Such non-compliance of USAR search markings by specialist teams has been observed in other disasters, such as the Canterbury 2011 earthquake [13,14].

According to one respondent “the Cajun Navy did a good job, but was a problem” and that this group, along with other organizations from outside the county, allegedly acted outside the emergency management system, illegally entered properties, took animals from flood-affected properties with no reunification plan, contributed to disease outbreaks (such as parvovirus) due to a lack of understanding local veterinary health issues, had no record keeping and took animals out of state never to be reunified with their owners. One respondent summed up their frustrations in saying “So in (Hurricane) Harvey if you were missing a pet it was with Harris County Animal Control, Houston SPCA, Houston Humane Society or stolen by some dude on a boat”. Yet another respondent said that the Cajun Navy had the benefit that it had “no red tape” meaning it flaunted the need for insurance or worry about jurisdictional issues or other restrictive policies. The issue of spontaneous animal volunteer groups often working outside the emergency management system and/or creating challenges in response has been well documented as an issue [5,7,15,16,17,18].

In a rural county, flood-displaced and evacuated horses were corralled and there were instances of people turning up purporting to be the owner (disaster rustling) as without microchipping or other forms of identification it was difficult to establish ownership of the animal. According to one respondent in this county’s large animal evacuation centre, a volunteer took the initiative of telling people purporting to be the owner reclaiming their horse “that if this was not their horse, it was a felony offence and he made sure he took a photo of them, the horse, and their identification so he could pass it onto law enforcement if required”.

Evacuation failure associated with animal ownership has been well researched. Though evacuation failure observations are outside the scope of this study there is strong evidence across the literature that the evacuation of companion animals alongside their human guardians positively affects public safety, including that of the animal owners and emergency responders [19,20,21,22,23,24,25,26]. Research on evacuation failure has been well articulated with Heath and Linnabary’s statement that “there is no other factor contributing as much to human evacuation failure in disasters that is under the control of emergency management when a threat is imminent as pet ownership” [27].

### 3.4. Media

The media portrayal of Hurricane Harvey was much more positive than that of Hurricane Katrina with an outpouring of public support rather than public outcry. Many public figures such as the US President Donald. J. Trump giving the Houston Humane Society a personal donation of US$25,000 [28], singer and songwriter Amanda Lambert, who co-founded the animal welfare charity Mutt Nation Foundation, responded to the affected area to help empty local animal shelters to create space for flood-displaced animals [29]; and even an Australian all-male review group based in Las Vegas were in town and volunteered their time bathing and walking flood-affected animals while also making a US$5000 donation to the Houston Humane Society [30].

Barnes et al. found that during and following Hurricane Katrina, the media were more likely to portray the efforts of individuals and non-profits in a more positive light, than the efforts of government and for-profit businesses [31]. Although, anecdotally, the overall media coverage during Hurricane Harvey was more positive than during Hurricane Katrina, the positive coverage of individual and non-profit group efforts appeared to, again, be given more attention than the issues or performance of government and for-profit organizations. 

### 3.5. Donated Goods

Hurricane Harvey demonstrated the generosity of Americans with an overwhelming deluge of donated goods to animal charities helping in the response. One respondent said, “thank God we have a warehouse” and “It’s a good thing, but it’s like another disaster” referring to the excessive volume of donated goods which also became a distraction to providing a response. At one point, one animal shelter had five to six FedEx trucks permanently cuing throughout the day to drop off donated goods for up to six days, leading to the police visiting due to the traffic congestion caused by the donations. In another example, a comment on social media asking for peanut butter for enrichment toys resulted in a major flood of donated peanut butter and the post was taken down.

The repeated experiences of excessive donations of goods that are often inappropriate, used, or expired are well-documented [27,32,33,34] and it would appear, again, as the lessons of the past have not been learned. This lack of lesson learning is not unique to Hurricane Harvey or the animal emergency management sector, as it is a challenge globally and there is yet to be a well-developed lesson learning system, as, at best, most current models are over-simplified and lack an evidence-based dynamic doctrine approach which allows for real-time incident management adjustment [35]. The World Organisation for Animal Health (OIE) guideline on disaster management also recommends that animal disaster management programmes incorporate a “lessons learned” system [36], though the term may be a misnomer given these are more likely to be a “lessons identified” system.

### 3.6. Service Animals

One of the key characteristics of the PETS Act because of the experiences in Hurricane Katrina was the specified inclusion of service animals, those used to assist people with disabilities. Respondents all reported that there were no known issues with service animals during Hurricane Katrina. However, service animal users were not interviewed, and further research would be needed to substantiate the assumption that there were no issues.

### 3.7. Reunification

Following Hurricane Harvey, the lack of standardized and centralized displaced animal forms and databases respectively for animal emergency management continue to be problematic as experienced over a decade since Hurricane Katrina. Microchipping is not common in Texas according to many respondents and the fragmented nature of animal groups in the US meant in the respondent’s opinion that the development of a standardized form or database for displaced/evacuated animal information was “never going to happen” as there are too many organizations that would have to agree. Again, the US is not alone with New Zealand also having faced the same animal information and data challenges in the Canterbury 2010 earthquake [37] and the Edgecumbe 2017 floods [12].

Animals that came into the care of one major animal welfare charity in Houston were held for 30 days, which was a variation of the normal legal requirement of three days. This was consistent with the American Bar Association’s [38] recommendation of 30 days post-disaster animal holding periods. Some other organizations did not extend their holding periods, with some only holding for ten days according to one respondent.

Post-disaster community clinics were provided by major animal welfare charities including free vaccinations, heartworm tests and microchipping. One such clinic expected 300 animals, but over 1500 animals were presented, requiring many to be given vouchers to come back another time.

One major animal welfare charity in Houston also launched “Operation Reunite” that enabled over 300 displaced animals to be placed in veterinary clinics as fosters instead of being housed in traditional temporary animal shelters; and developed a webpage “that would allow pet owners needing a temporary foster home for their pets to connect with potential foster homes and select the best fit for them” [24].

### 3.8. Other Legal Issues

In addition to disaster rustling, illegal rescues and animal holding periods, there were many other legal observations. Of those respondents that were interviewed, many had a general or animal-specific law enforcement role, directly or indirectly as part of their organisation’s mandate. Many confirmed that there were acts of abandonment that required to be investigated. However, the use of the Texas Health and Safety Code [39] appeared effective, specifically section 821.077 that prohibited the “unlawful restraint of a dog”. This Texan code makes it illegal to tether a dog outside in extreme weather or when a Hurricane warning is in effect. This was reflected by Roman City Forest Police Chief Stephen Carlisle gaining the attention of international media by quoting “I promise you that I will hold anyone accountable that unlawfully restrains their dog” [40] in the lead up to Hurricane Harvey.

One animal welfare charity who undertakes humane law enforcement also under-covered a major animal hoarding abuse case. Though the method of detecting the offending has been kept in confidence to preserve the pending investigation, it would appear from an interview with a respondent involved in the investigation that animal hoarders used the disaster as an opportunity to re-stock their animal numbers. With the urgent need to clear existing animal levels within animal shelters to make room for disaster-displaced animals, the abundance of displaced animals and the over-supply of donated cages available to the public to assist with evacuations, these conditions are ripe for disaster hoarding to occur. It may well be that Hurricane Harvey has exposed the first empirical case of disaster hoarding which may further highlight the need for animal law enforcement agencies to strategically prioritise animal emergency management as a core function.

## 4. Future Work

The limited number of respondents interviewed may limit the findings of this study. Further research is needed on a larger scale to survey a wider sample of animal emergency responders, including those from out of state and those who were from spontaneous volunteer groups, such as the Cajun Navy.

The potentially novel case of observed disaster hoarding along with other acts of disaster animal abuse may be indicative of crimes that have traditionally been out of scope or view for researchers and the interaction between disasters and animal abuse may create new sub-disciplines within the constantly-evolving niche subject of animal emergency management. 

## 5. Conclusions

In the twelve years since the introduction of the PETS Act, the United States has culturally made the preservation of companion animals in disasters a priority. Though there is some evidence to suggest the PETS Act has contributed to this cultural change, the implementation of animal emergency planning appears sub-optimal and the integration of animal welfare charities to respond effectively remains fragmented in many areas. Hurricane Harvey repeated many of the challenges observed in previous emergency events including Hurricane Katrina, from overwhelming donations of goods, lack of coordination, unreasonable abandonment, lack of common reunification systems, inter-organizational tensions, and lack of preparedness. This, however, is no different from many other countries, including New Zealand, and this reinforces the need for improved lesson learning systems and for the animal emergency management community to collaborate more on an international level.

## Figures and Tables

**Figure 1 animals-08-00047-f001:**
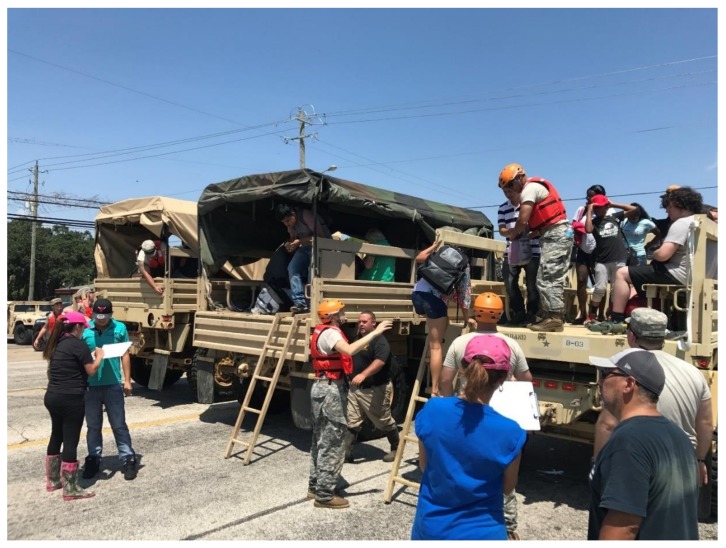
US military assists with appropriate protective equipment for flood response during the Hurricane Harvey response (photo credit: Wharton City Police Department).

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
