# Peer review of "Did Harvey Learn from Katrina? Initial Observations of the Response to Companion Animals during Hurricane Harvey"

_animals, 2018, doi:10.3390/ani8040047_

Round 1

Reviewer 1 Report

Did Harvey learn from Katrina? Initial observations 2 of the response to companion animals during 3 Hurricane Harvey

Well written and a good balanced presentation of an important tropic. The discussion could be enriched by looking at two overlooked references

Heath SE, Kass PH, Beck AM, Glickman LT. Human and pet-related risk factors for household evacuation failure during a natural disaster. American Journal of Epidemiology. 2001 Apr 1; 153(7):659-65.

Heath SE, Beck AM, Kass PH, Glickman LT. Risk factors for pet evacuation failure after a slow-onset disaster. Journal of the American Veterinary Medical Association. 2001 Jun 1; 218(12):1905-10.

Some basic comments:

Line

10        typo ‘2015’ should be ‘2005’ and “to better protection” should be?

11        ‘2016’ should be ‘2006’

101-106 Very important points and could be emphasized

109 type ‘countries’ should be ‘counties’

190 authors should check

Barnes MD, Hanson CL, Novilla LM, Meacham AT, McIntyre E, Erickson BC. Analysis of media agenda setting during and after Hurricane Katrina: Implications for emergency preparedness, disaster response, and disaster policy. American Journal of Public Health. 2008 Apr; 98(4):604-10.

253      Hoarding a new insight, good

272      Conclusion very good

Author Response

Firstly, thank you for taking the time to review the article. It would well appear that you are very familiar with this area of research and you have made excellent feedback, all of which I have now made.

Well written and a good balanced presentation of an important tropic. The discussion could be enriched by looking at two overlooked references

Heath SE, Kass PH, Beck AM, Glickman LT. Human and pet-related risk factors for household evacuation failure during a natural disaster. American Journal of Epidemiology. 2001 Apr 1; 153(7):659-65.

Heath SE, Beck AM, Kass PH, Glickman LT. Risk factors for pet evacuation failure after a slow-onset disaster. Journal of the American Veterinary Medical Association. 2001 Jun 1; 218(12):1905-10.

Both these added along with discussion around evacuation failure.

Evacuation failure associated with animal ownership has been well researched. Though evacuation failure observations are outside the scope of this study there is strong evidence across the literature that the evacuation of companion animals alongside their human guardians positively affects public safety, including that of the animal owners and emergency responders [19–26]. Research on evacuation failure has been well articulated with Heath and Linnabary’s statement that “there is no other factor contributing as much to human evacuation failure in disasters that is under the control of emergency management when a threat is imminent as pet ownership” [27].

Additional references as suggested are added in discussion and cited. I am constrained the discussion of evacuation failure (which was a well pointed out glaring omission on my behalf) as the other reviewer already wanted the article to be condensed, which I don't think I can really do with already some sections only a paragraph or two, and could be further expanded upon.

Some basic comments:

Line

10        typo ‘2015’ should be ‘2005’ and “to better protection” should be?

…to better afford protection…

11        ‘2016’ should be ‘2006’

         Corrected

101-106 Very important points and could be emphasized

Statement added: “This potentially exposes a major shortfall in the US animal emergency management environment”.

109 type ‘countries’ should be ‘counties’

         Corrected

190 authors should check

Barnes MD, Hanson CL, Novilla LM, Meacham AT, McIntyre E, Erickson BC. Analysis of media agenda setting during and after Hurricane Katrina: Implications for emergency preparedness, disaster response, and disaster policy. American Journal of Public Health. 2008 Apr; 98(4):604-10.

         Thank you, this was an article I was unaware of. New point added with citation.

         Barnes et. al. found that during and following Hurricane Katrina, the media were more likely to portray the efforts of individuals and non-profits in a more positive light, than the efforts of government and for-profit businesses [31].  

253      Hoarding a new insight, good

         Thank you

272      Conclusion very good

         Thank you again for your review.

Reviewer 2 Report

This paper should be of interest to those concerned with disaster preparedness and relief for both people and their animal companions. However, it has typographical mistakes, errors in dates, and some odd verbiage. The text can be tightened up and the paper shortened. 

Specific Comments: Lines 10 -11.  2005;  "offer better protection of companion animals, " of 2006.  Line 29. Delete "potentially". Line 40. Insert "of "2006. line 42. "330,000". Line 44.  There "were" particularly -- . Line 48. global media, documented one community-centric --- . Line 53. considerably "lower ---. Lines 63-66. Awkward sentence.Lines 74-75. How "many" also received follow up emails, when only six subjects were interviewed ? Line 81. -- nature "of the interview" --. Line 86 "were" and Line 88. "were". Line 109. "counties" . Line 140. "were". Line 151. Suggest  The US "military were highly praised from all the subjects"---. Line 170.  Delete "noted".  Line 173. parvovirus (lower case). . Line 174. "allegedly" took animals out of state. Line 182, "coralled" . Line 183. Delete "saying".   Lines 185-186. Awkward sentence. Line 194. Delete "work". Line 200. Capitalize "God".   Lines 208-212. Suggest making a new sentence here at line 210. -- system. At best, most -- . Line 224. Insert "that" the development of a ----.  Line 226. "alone" . Line 230.  --a variation "of" the normal ---.   Line 247. "effective".   Lines 248-249. Awkward wording.  Line 273. In the "12" years since the ----.

Author Response

Thank you very much for reviewing this paper.

This paper should be of interest to those concerned with disaster preparedness and relief for both people and their animal companions. However, it has typographical mistakes, errors in dates, and some odd verbiage. The text can be tightened up and the paper shortened. 

It has been a nice balance of feedback from yourself and the other review. The other reviewer had some very good technical recommendations, all of which I have now incorporated.

I am reluctant to reduce the length of the article (tighten up) given there are many aspects that could be further expanded upon, hence the initial observations nature of the article. Some additional content was added to address suggestions made by the other reviewer. I am assuming your comments are directed to address the rating scores given, and if so, I have no worked through all your comments and rectified these. Having this level of scrutiny has been very useful to ensure the article is fit for publication (I wish I knew someone like you to pre-review my articles prior to submission!).

Specific Comments:

Lines 10 -11.  2005;  corrected

"offer afford better protection of companion animals… corrected

" of 2006.  corrected

Line 29. Delete "potentially". corrected

Line 40. Insert "of "2006. corrected

 line 42. "330,000". corrected

Line 44.  There "were" particularly -- . corrected

Line 48. global media, documented one community-centric --- . corrected

Line 53. considerably "lower ---. corrected

Lines 63-66. Awkward sentence.

Lines 74-75. How "many" also received follow up emails, when only six subjects were interviewed ? All, corrected.

       Six subjects were interviewed and all received follow up emails to clarify notes taken and request feedback on the final manuscript prior to peer review.

Line 81. -- nature "of the interview" --. corrected

Line 86 "were" corrected

Line 88. "were". corrected

Line 109. "counties" . corrected

Line 140. "were". corrected

Line 151. Suggest  The US "military were highly praised from all the subjects"--- corrected.

       The US military were highly praised by all the subjects for their response efforts including the rescuing of animals.

 Line 170.  Delete "noted".  corrected

Line 173. parvovirus (lower case). . corrected

Line 174. "allegedly" took animals out of state. corrected

Line 182, "coralled" . corrected

Line 183. Delete "saying".   corrected

Lines 185-186. Awkward sentence. Revised.

       In a rural county, flood-displaced and evacuated horses were coralled and there were instances of people turning up purporting to be the owner (disaster rustling) as without microchipping or other forms of identification it was hard to establish ownership of the animal. According to one respondent in this county’s large animal evacuation centre, a volunteer took the initiative of telling  people purporting to be the owner reclaiming their horse “that if this was not their horse, it was a felony offence and he made sure he took a photo of them, the horse and their identification so he could pass it onto law enforcement if required”.

Line 194. Delete "work". corrected

Line 200. Capitalize "God".   corrected

Lines 208-212. Suggest making a new sentence here at line 210. -- system. At best, most -- corrected

Line 224. Insert "that" the development of a ----.  corrected

Line 226. "alone" . Line 230.  --a variation "of" the normal ---.   corrected

Line 247. "effective".   corrected

Lines 248-249. Awkward wording.  Revised

However, the use of the Texas Health & Safety Code [39] appeared effective, specifically section 821.077 that prohibits the unlawful restraint of a dog. This Texan code makes it illegal to tether a dog outside in extreme weather or when a Hurricane warning was in effect.

Line 273. In the "12" years since the ----. corrected

Your feedback has added great value to the paper and I hope the corrections made, now meet your approval to publish.

Kind regards

Author

Round 2

Reviewer 2 Report

 The revised manuscript is much improved, Glad that the author understood the points raised.